# Controlled growth of single-crystalline metal nanowires via thermomigration across a nanoscale junction

De-Gang Xie[1], Zhi-Yu Nie[1], Shuhei Shinzato[2], Yue-Qing Yang [1], Feng-Xian Liu[3], Shigenobu Ogata [2,4]*, Ju Li [1,5]*, Evan Ma [1,6] & Zhi-Wei Shan [1]*

Mass transport driven by temperature gradient is commonly seen in fluids. However, here we demonstrate that when drawing a cold nano-tip off a hot solid substrate, thermomigration can be so rampant that it can be exploited for producing single-crystalline aluminum, copper, silver and tin nanowires. This demonstrates that in nanoscale objects, solids can mimic liquids in rapid morphological changes, by virtue of fast surface diffusion across short distances. During uniform growth, a thin neck-shaped ligament containing a grain boundary (GB) usually forms between the hot and the cold ends, sustaining an extremely high temperature gradient that should have driven even larger mass flux, if not counteracted by the relative sluggishness of plating into the GB and the resulting back stress. This GB-containing ligament is quite robust and can adapt to varying drawing directions and velocities, imparting good controllability to the nanowire growth in a manner akin to Czochralski crystal growth.

[1] Center for Advancing Materials Performance from the Nanoscale (CAMP-Nano) & Hysitron Applied Research Center in China (HARCC), State Key Laboratory for Mechanical Behavior of Materials, Xi'an Jiaotong University, Xi'an 710049, China. [2] Department of Mechanical Science and Bioengineering, Osaka University, Osaka 560-8531, Japan. [3] Applied Mechanics Lab., School of Aerospace Engineering, Tsinghua University, Beijing 100084, China. [4] Center for Elements Strategy Initiative for Structural Materials (ESISM), Kyoto University, Kyoto 606-8501, Japan. [5] Department of Nuclear Science and Engineering and Department of Materials Science and Engineering, Massachusetts Institute of Technology, Cambridge, MA 02139, USA. [6] Department of Materials Science and Engineering, Johns Hopkins University, Baltimore, MD 21218, USA. *email: ogata@me.es.osaka-u.ac.jp; liju@mit.edu; zwshan@mail.xjtu.edu.cn

Mass transport driven by the temperature gradient is well known in gases and liquids, as fluids can readily undergo mass transport via convection or Soret diffusion[1,2]. In contrast, in solids, the temperature gradient rarely builds up to a level high enough to drive thermomigration (one of Onsager's famous off-diagonal linear responses) to cause a rapid shape change[3]. However, for nanomaterials, thermomigration could become a powerful mechanism for growth or shape change. When the characteristic dimension decreases to the nanoscale, the surface/bulk ratio renders surface diffusion dominant over other diffusion mechanisms. In fact, surface transport is so fast that the material can plastically deform via purely diffusional mechanisms[4,5]. The characteristic length scale for surface diffusional mechanisms to dominate, $L_s$, is related to the homologous temperature ($\equiv$ temperature $T$/bulk melting temperature $T_m$) of the material. For example, at room temperature and a typical strain rate like $10^{-2}$/s, $L_s = $ ~200 nm for Sn[5], but ~10 nm for Ag[4]. Below $L_s$, surface diffusion-mediated deformation can happen on the timescale of seconds to minutes[5]. Moreover, as the driving force, the temperature gradient in a nanostructure can reach a level much higher than that in the bulk counterpart, because the thermal resistance $R \propto L/A$, where $L$ and $A$ are the length and cross-sectional area of a structure, respectively, and this is especially true for one-dimensional structures like nanowires, or nanoligaments[6]. For the reasons above, thermomigration can happen at the nanoscale to quickly reshape a nanostructure. While grain-boundary (GB) diffusion is similar to surface diffusion and often acts cooperatively to accommodate the thermomigration atomic current, by first transporting along and later depositing/stripping into GB as the sink/source[7], here, we do want to make a distinction between surface and GB diffusions in that the latter process can be significantly more sluggish than the former depending on $T/T_m$ due to the somewhat lower free volume inside GB than the free surface, so much so that the GB process can be the bottleneck when the two processes are coupled. Unlike a free surface, atoms in GB are sandwiched by two solid bodies, and are able to generate significant back stress normal to the GB when there is traffic jam inside GB[8]. This distinction turns out to be important for the thermomigration nanostructure with both free surfaces and GBs.

The question is then how to harness the extraordinary thermomigration at the nanoscale. The fast shape change can be both harmful and useful depending on the context. For integrated circuits, thermomigration has been considered as a threat to structural reliability of nanoscale interconnects[9,10] with occasional temperature gradient typically <1000 °C cm$^{-1}$. But if well harnessed, thermomigration can be potentially useful[11–14], in applications such as reshaping or growing nanostructures. This is similar to pulling single crystals out of liquids in Czochralski growth[15,16], except with surface thermomigration one has a 2D liquid layer—the surface premelting layer covering the solid reservoir—instead of a 3D liquid bath beneath. Also, unlike displacive deformation, after diffusive deformation the material usually remains a clean crystal with defect-free interior, making it a more desirable way of nanoscale reshaping.

In this work, we propose and demonstrate a thermomigration-based method to controllably grow metallic nanowires such as Al, Cu, Ag, and Sn, directly from the surface of a hot solid by simply drawing a cold tip back after touching. Our proposed method combines the advantages of the traditional Czochralski method[15,16] of pulling single crystal out of liquid and conventional metalworking process of wire drawing, to make single-crystal nanowires directly from a solid reservoir without a holed die. Previously, it has been demonstrated that diffusion from free surface into 1D dislocation and GB can lead to superplasticity[17,18], but a Czochralski type

growth (with a GB replacing the liquid–crystal interface) is unprecedented.

## Results

**In situ TEM experiments of making nanowires.** The proposed method for growing metallic nanowires is illustrated in Fig. 1. In this method, three preconditions are required: The first is a hot solid reservoir free of confinement from surface oxide[19], which can be attained by scratching the pulling tip on the hot aluminum substrate to break the native oxide layer and expose fresh metal; The second is the sharp temperature gradient to induce thermomigration, which is achieved by touching a cold nanoscale tip with the hot metal reservoir: this nucleates a small seed that often has a different crystal orientation from the hot metal reservoir beneath; The third one is the mechanical pulling movement to enlarge the seed, and later sustain it as steady-state nanowire growth, with a neck-shaped region bridging the nanowire and the substrate. Again, we emphasize that the metal grown does not inherit the crystal orientation of the hot substrate beneath, and within the neck region there exists a grain boundary (GB); Plating into the GB feeds the growth of the upper, colder crystal, as the newly arrived atoms choose to deposit onto the colder side of the GB and take the lattice orientation of the new crystal. This allows the new crystal to grow taller and taller. Simultaneously, there are curvature-driven[20] (and later stress driven[21]) surface diffusional fluxes in the neck region as well, to maintain steady-state neck ligament shape.

To verify the proposed method above, we conducted experiments by mounting a metal foil onto a miniaturized heater at the front end of the TEM holder (Fig. 2a). The opposite side is a movable diamond tip or tungsten tip as the cold end to realize the drawing operation. At drawing speed <10 nm s$^{-1}$, nanowires can be drawn out from a hot metal substrate held at temperature >0.5$T_m$. Examples of making Al, Cu, Ag, and Sn nanowires were shown in Fig. 2 (see also Supplementary Movie 1), Supplementary Note 1 and Supplementary Figs. 1–3. The making of Al nanowires is chosen as the main example, while others are used to demonstrate the transferability of the method. The reason is that aluminum has a relatively low $T_m$ (933 K), so $T/T_m$ can be as high as 0.72 by applying a heater. Another reason for choosing aluminum is that high-quality nanowires of active metals like aluminum remain difficult to make with existing synthetic methods[22–26]. Template-assisted processes[27–29], the most frequently used method of making aluminum nanowire, usually produce nanowires that are corroded, contaminated, and bundled together.

After heating up the aluminum to 400 °C, a cold tip was manipulated to touch the hot surface of aluminum and then pulled up slowly. A small bump on the substrate was chosen to be the initial contact site, as it allows easy observation of the subsequent mass transport: the mass in the bump was sucked up, as indicated by the recession of its profile from Fig. 2b to Fig. 2c, d. After 520 s, a rough nanowire 1.5 μm in length and 100 nm in diameter was formed. This nanowire consists of several segments, with dents on the surface and GBs inside the wire. This polycrystalline growth is because the positioning system used here could move stably only across a small distance of a few hundred nanometers. The nanowire growth is easily perturbed by vibrations or lateral drift, causing the change of growth direction or surface roughness.

However, from 520 to 691 s, a thin nanowire with uniform diameter was produced, as shown in Fig. 2e–g. During this stage of uniform growth, the region at the bottom part of the nanowire is boxed in the red rectangle, which is enlarged to pixel resolution in the inset in Fig. 2e. The pixel measurement results indicate that

 

a neck-shaped ligament, with characteristic height $h \sim 50$ nm and an apparent diameter of $\Phi_n = 7$ nm in its narrowest part, was formed, whereas the uniform part of nanowire has a much larger apparent diameter $\Phi_w = 40$ nm.

Despite its much smaller size, the thin neck-shaped ligament can tolerate significant level of tip shake, thus can help stabilize the growth of the nanowire. Although slight shaking still occurred to the drawing tip, the ensuing growth process is quite stable (see Supplementary Movie 1), as long as the neck-shaped ligament still sits at the bottom, as schematically depicted in Fig. 1. The balance of several competing factors determines the steady-state neck shape. One is obviously the temperature distribution and the divergence of the thermomigration flux $J_T \propto -D_s \nabla T$, where $D_s$ is

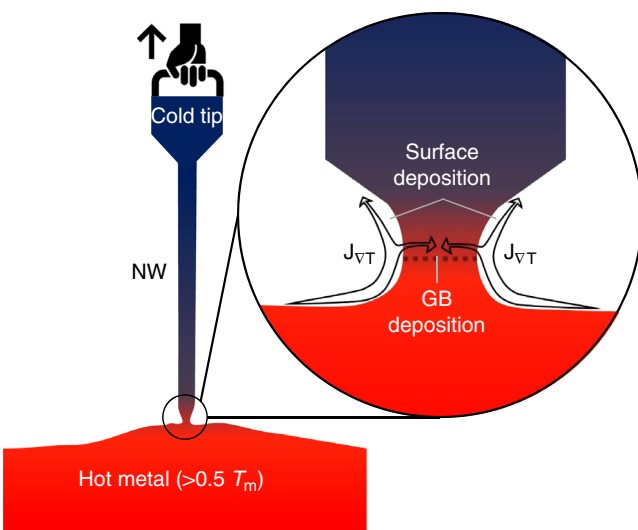

**Fig. 1** Schematic illustration of making single-crystalline nanowire by hot drawing. The physical process occurring near the neck-shaped ligament with a grain boundary (GB) is shown in the magnified insert

the surface diffusivity. One is the local pulling load[21,30], which tends to narrow the neck; but the load is itself dependent on the pulling speed and the ease of GB diffusion/plating relative to the rampant surface diffusion (if there were just surface diffusion but no GB diffusion/plating, the nanowire can grow fatter but not taller, and the load cannot be relieved). Yet another factor is the surface curvature-driven smoothening process[20,31].

The nanowire growth terminates when there is an abrupt breakup of the neck, as shown in Fig. 2f. The lateral drift of the drawing tip gradually bent the nanowire, exerting an additional shear load to the thin neck and resulting in the ultimate breakup. A comparison of the nanowire profile is shown in Fig. 2g, revealing the elastic relaxation of the nanowire back to a straight shape. The breakup of the neck and the elastic spring-back of the nanowire can be used to estimate the shear strength of the neck. The shear strength is estimated to be ~300 MPa, indicating that the neck is still strong solid (see Supplementary Note 2).

Since the nanowire was mechanically pulled from the hot solid substrate like Czochralski growth, it raises a question as to how much pulling stress exists in the neck and nanowire to meet the required tip displacement rate (growth rate). To clarify this point, we adopted a high-sensitivity force transducer to measure the pulling load. The result is shown in Supplementary Note 3 and Supplementary Fig. 4, which demonstrates that the uniaxial tensile stress felt by the nanowire was only a few tens of MPa. This is much smaller than the stress required to activate dislocation plasticity in such nanowires, thus proving that dislocation plasticity are not necessary ingredients for the observed growth. Also, possible electron-beam effect on nanowire growth was also excluded, as detailed in Supplementary Note 4 and Supplementary Fig. 5. Therefore, the nanowire growth in our work was not a conventional wire drawing process.

Thermomigration occurs along temperature gradient. Since direct measurement of the nanoscale temperature distribution is still a formidable challenge, we resort to an estimation of the temperature gradient based on the dynamic measurement of nanowire growth. Figure 2e–f show the nanowire in the uniform

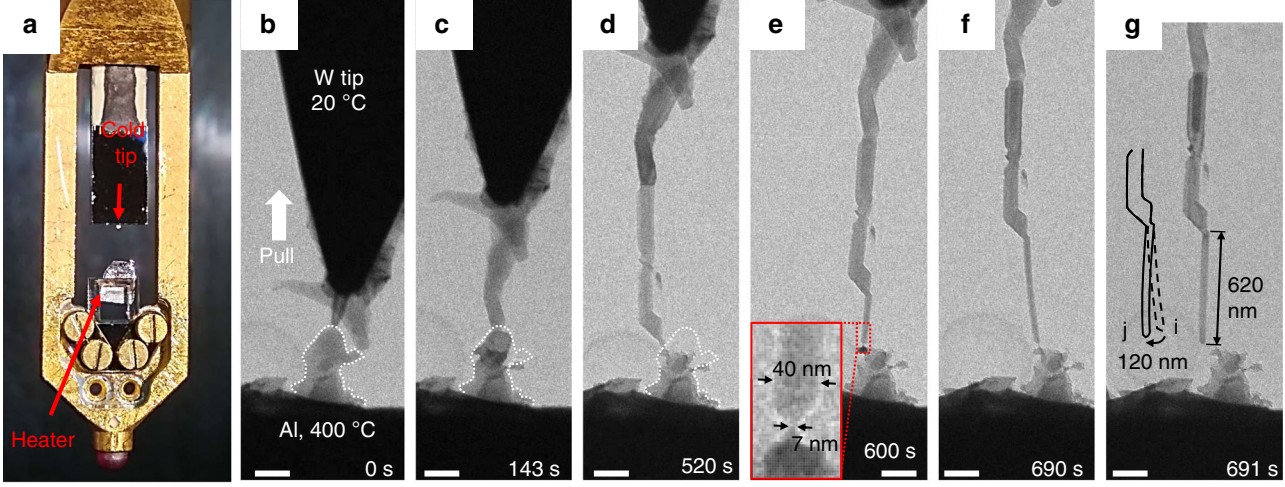

**Fig. 2** The experimental setup and the process of making an aluminum nanowire by drawing from an Al hot solid substrate. **a** The front end of the TEM sample holder is mounted with a miniature resistive heater, opposite to which is a cold movable tip. **b–d** After heating the aluminum substrate to 400 °C, the cold W tip was brought to contact with the hot substrate. Upon pulling, an aluminum nanowire grows out between the tip and substrate. The profile of the substrate near the contact point is depicted with the dashed white line and superimposed to (**c**) and (**d**), indicating that the mass was sucked up to build the nanowire. **e** When a slim neck-shaped ligament formed between the nanowire and substrate, uniform growth of the nanowire was observed to ensue. The characteristic diameter of the neck and the nanowire is measured to be 7 and 40 nm, respectively. **f**, **g** With the pulling tip ascending, the nanowire grew longer to 620 nm, until the lateral tip shift bent the nanowire to breakup. The inset sketch compares the nanowire profile before and after relaxation. Scale bars in (**b–g**) represent 200 nm

 

growth stage, with a nearly constant velocity of $v = 3.8 \, \text{nm s}^{-1}$ (see Supplementary Fig. 6). With this velocity and the nanowire geometry, the atomic flux ($J_s$) via surface diffusion[4] can be calculated as

$$J_s = \frac{dV}{dt} \frac{1}{\Omega A_n} = \frac{\nu A_w}{\Omega A_n} = \frac{\nu \Phi_w^2}{4\Omega \Phi_n \delta_s} \qquad (1)$$

where $\Omega$ is the atomic volume of Al ($1.66 \times 10^{-29} \, \text{m}^3$), $A_n$ is the cross-sectional area of the surface pre-melting layer at the neck with a nominal surface layer thickness $\delta_s = 3 \, \text{Å}$, $A_w$ is the cross-section area of the nanowire. The resultant $J_s = 4.5 \times 10^{22} \, \text{m}^{-2} \, \text{s}^{-1}$.

For elemental metals, the thermomigration flux contribution $J_T$[32,33] is written as

$$J_T = -\frac{Q D_s}{\Omega k_B T^2} \nabla T \qquad (2)$$

where $Q$ is the coefficient of heat transfer ($Q = 0.07 \, \text{eV}$ in the bulk aluminum single crystal)[34], $k_B$ is the Boltzmann constant, $T$ is the average temperature across the neck, and $\nabla T$ is the temperature gradient. Equation (2) is called an off-diagonal term in the Onsager linear response theory: while it is well known that mass flux can be driven by chemical potential gradient $\nabla \mu$ and heat flux by temperature gradient $\nabla T$ (Fick's law and Fourier's law, the on-diagonal terms), $\nabla T$ can also drive mass flux (thermomigration) and reciprocally $\nabla \mu$ can also drive heat flux, as surface atoms are both heat and mass carriers. Suppose $J_s = J_T$, that is, suppose $\nabla \mu$ somehow does not contribute, the detailed estimation of $\nabla T$ is described in Supplementary Note 5 and Supplementary Fig. 7. This calculation showed that the requisite temperature difference to sustain the nanowire growth velocity would only need to be 0.05 K, if the overall growth velocity is long-range thermomigration/surface diffusion controlled instead of interfacial reaction (i.e., GB diffusion/deposition) controlled, and $\nabla \mu$ does not kick in as feedback. This should be the minimum required temperature difference, which corresponds to $(\nabla T)_{\min} = 1.3 \times 10^6 \, \text{K m}^{-1}$ away from GB. We then performed finite element modeling (FEM) by trying different thermal conductivity ($\kappa$) values in the neck region, as shown in Supplementary Fig. 10b, c, with/without off-diagonal contribution to the heat flux from mass flux. These parametric simulations show that the real temperature difference cannot be this small, and the real temperature gradient must be much higher than $(\nabla T)_{\min} = 1.3 \times 10^6 \, \text{K m}^{-1}$.

To estimate the real temperature distribution, inputting physically plausible $\kappa$ in the neck ligament and nanowire is critical, which has to consider two effects, the contribution from the surface mass transport and the quantum size effect. First, we estimated the effective additional thermal conductivity from surface mass transport, as described in Supplementary Note 6 and Supplementary Fig. 8. The results show that the extra heat conduction from off-diagonal contribution is at least four magnitudes smaller than the on-diagonal bulk value, so that its effect is numerically negligible. For the quantum size effect, the thermal conductivity has a strong size effect at the nanoscale regime, when the geometrical size approaches the electron mean free path ($\lambda_e = \sim 22 \, \text{nm}$ for Al near RT)[35] or the phonon mean free path ($\lambda_{ph} = \sim 5$–$7 \, \text{nm}$ for Al near RT)[36,37]. The size of nanowire and the neck ligament is close to $\lambda_{ph}$ or $\lambda_e$, so that the thermal conductivity in both the neck and the nanowire can be much smaller than the bulk value[38,39]. By using extrapolation shown in Supplementary Note 7 and Supplementary Fig. 9, thermal conductivity at the neck and nanowire was estimated to be 40 and 75 $\text{W m}^{-1} \, \text{K}^{-1}$ (compared with bulk value of $\kappa > 200 \, \text{W m}^{-1} \, \text{K}^{-1}$). Based on these modified values, we performed FEM modeling and the results are detailed in Supplementary Note 8 and Supplementary Fig. 10. From the FEM results, the real temperature gradient must be as high as $10^{10} \, \text{K m}^{-1}$. Thus, the

real thermomigration driving force is about four orders of magnitude larger than the what it needs to be, if it runs unopposed ($\nabla \mu$ term silent).

This result indicates there must exist some other factors from $\nabla \mu$ term to counter-balance the thermomigration, such as mass back flow driven by back stress and/or surface curvature, a phenomenon noticed as a side effect during thermomigration in interconnects of microchips. Specifically, a mechanical back-stress $\sigma$ can be generated at the depositing sites, i.e., inside the GB within the neck. Since the GB diffusion/deposition is much slower than free surface diffusion, back stress generated in the GB can stifle the overall growth rate. Because there is a $\sigma \Omega$ term in the chemical potential, the thermomigration driving force can be completely canceled out if

$$\Omega \Delta \sigma = Q (\Delta \ln T) \qquad (3)$$

Using the temperature difference across the neck from FEM simulation, $\sim 400 \, ^\circ\text{C}$ to $\sim 200 \, ^\circ\text{C}$, a back-stress difference of 134 MPa can thus completely cancel the thermomigration flux in regions away from the GB. The logic here is that if GB diffusion/deposition is very sluggish (interfacial reaction controlled overall kinetics), but is nonetheless required to match the surface diffusion influx at steady-state, what will happen is that stress $\sigma$ will be auto-generated in the GB, that is telegraphed to the rest of the solid surfaces to slow down the surface influx. A detailed mathematical treatment coupling $\sigma(\mathbf{x})$, surface curvature, $\mu(\mathbf{x})$, $T(\mathbf{x})$, and the imposed displacement rate $v$ is beyond the scope of the present paper. But we know such a level of back stress is not uncommon, as a similar magnitude was found before in study on electro-migration of interconnects[40,41], where plating into the GB is sluggish and hence rate-limiting step. The $\sim 100 \, \text{MPa}$ back-stress difference generated inside the neck region thus regulates thermomigration away from the neck in a stiff negative feedback. The discrepancy in $\nabla T$ above, by as much as four orders of magnitude, indicates that 99.99% of the thermomigration driving force (off-diagonal term) is canceled out by the auto-generated back-stress and/or surface curvature effect on $\nabla \mu$ (on-diagonal Fickian diffusion term). The estimated $\sim 100 \, \text{MPa}$ normal stress at the narrowest section $\Phi_n = 7 \, \text{nm}$ is consistent with the estimated stress magnitude of a few tens of MPa at the uniform section $\Phi_w = 40 \, \text{nm}$ in Supplementary Fig. 6.

**Control of nanowire shape.** The surface morphology of the nanowire can be controlled by changing the pulling direction, as shown in Fig. 3 and Supplementary Movie 2. The initial pulling direction is about 26° away from a <110> direction of the nanowire, and the obtained nanowire developed a saw-tooth-shaped surface, as shown in Fig. 3a–c and Supplementary Movie 2. However, when the pulling direction is $\sim 4°$ away from the same <110> direction, the nanowire can almost maintain a straight and smooth morphology, except for a few small surface steps, as shown in Fig. 3d, e and Supplementary Movie 3. Thus, it can be concluded that pulling along a low-index direction of the nanowire tends to produce a straight prismatic nanowire. Figure 4f is the dark-field image of the nanowire taken with $[\bar{1}\bar{1}1]$ diffraction spot, showing that the whole nanowire lights up as a single crystal and clear thickness fringes appears. The growth direction of the straight segment is always <110>, as shown in Supplementary Fig. 11. The substrate does not seem to have an obvious effect. These results indicate that despite the switch in pulling direction, the final nanowire still maintains a single crystal with pristine interior.

Moreover, two other interesting observations are important for revealing the nanowire growth mechanism. The first one is from the comparison of the aluminum tooth shape shown in Fig. 3b, c.

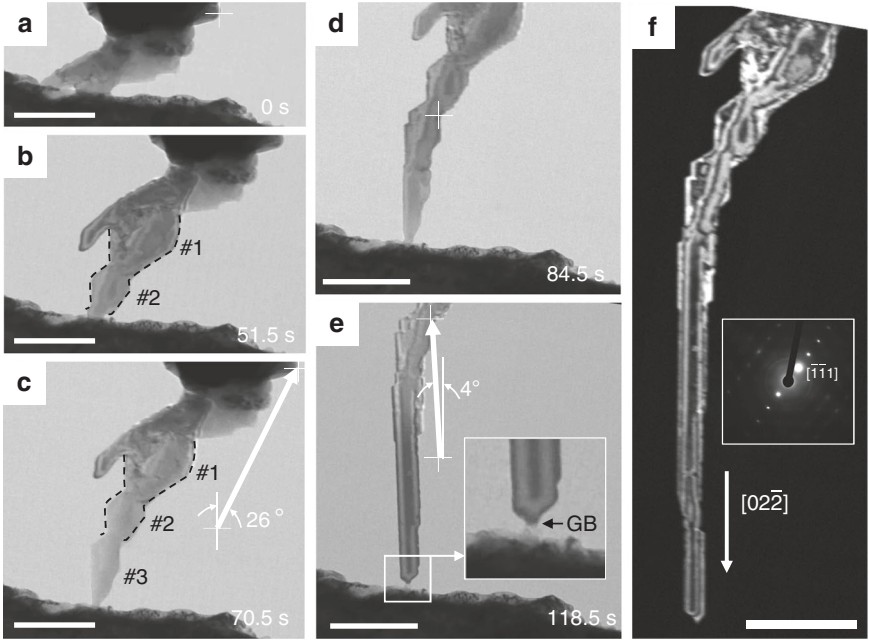

**Fig. 3** The nanowire growth for different pulling direction. **a–c** When pulling the tip towards the top right, the surface of the nanowire assumes a saw-tooth shape. A white cross with vertical line parallel to the $[02\bar{2}]$ direction of the nanowire marks the track of the movement of the pulling tip. The contour of two aluminum teeth is delineated with dashed black line and superimposed onto (**c**) for comparison. **d, e** When the pulling direction is upward, the nanowire morphology is relatively smooth. Inset in (**d**) shows a grain boundary (GB) in the neck region. **f** Dark-field image of the formed nanowire indicates single-crystalline internal structure. Inset is the diffraction pattern of the nanowire. Scale bars, 500 nm

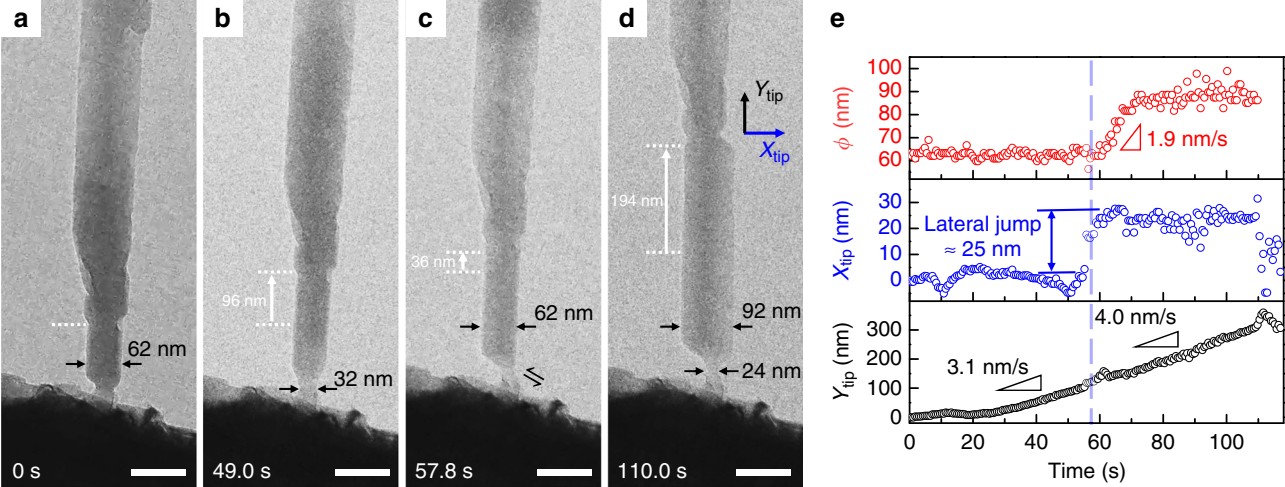

**Fig. 4** Tip movement to control the nanowire growth. **a, b** During the period of uniform growth, the length of nanowire increased by 96 nm while the apparent diameter (∅) remained unchanged. **c, d** at 57.8 s, a lateral jump of ~25 nm leftward occurred to the tip. After the jump, the nanowire showed increase in both length and ∅. **e** Plot of ∅ and the change of tip position in the pulling direction ($Y_{tip}$) and lateral direction ($X_{tip}$) as a function of time. Scale bars, 100 nm

After developing the shape as depicted by the dashed black line at 51.5 s in Fig. 3b, tooth #1 and #2 did not show visible growth from 51.5 to 70.5 s, while in this period, tooth #3 nucleated and grew up to a size comparable to #2. This suggests that tooth formation would completely stop once a new dent is formed between two teeth, i.e., the mass diffusion will not go across the dent. Actually, each tooth is a crystal starting its growth from the dented junction to inherit the lattice coherency. Second, a GB is observed inside the neck, as shown in the enlarged image in the inset of Fig. 3e, clearly visible only when the crystal is suitably oriented. Its existence not only reconciles the lattice

misorientation between the hot reservoir crystal and the newly formed crystal seeded by the cold tip, but also provide a diffusion path into the neck and atomic sink, such that deposition inside the neck can be realized, similar to the case with Sn[5]. The kinetic barrier of plating into this GB sink (vis-a-vis that of surface diffusion) likely controls whether the overall wire growth kinetics is surface transport controlled or interfacial reaction controlled[42,43]. Because our quantitative estimates indicate that major portion of the long-range transport driving force is counteracted by the back-stress gradient, and the back stress can be generated by inefficient GB sink actions that directly give

volume change normal to the GB (put it another way, a stress normal to the GB drives GB plating/stripping, the original argument made by Herring[44]), we conclude our wire growth kinetics is interfacial reaction controlled, that is, the great majority of the thermomigration driving force is spent on driving the GB plating reaction via a self-regulated back stress normal to the GB. This is quite reasonable, because GB diffusion is significantly more sluggish than surface diffusion at these temperatures due to the smaller free volume inside GB, and when surface diffusion is serially coupled to GB diffusion/reaction, it is the latter that is likely to be rate-controlling.

**Control of nanowire size**. Besides the control of nanowire morphology via drawing direction, the size of the nanowire is also tuneable by manipulating the relative position of neck-shaped ligament to the nanowire bottom. During the uniform growth, the neck-shaped ligament sits at the geometrical center of the nanowire cross section. If the ligament is displaced off the center by shearing the pulling tip, the uniform growth can be changed into non-uniform growth. This suggests that the ligament could be manipulated to tune the size/morphology of the nanowire, as shown in Fig. 4 and Supplementary Movie 4. In the first stage from 0 to 57.8 s in Fig. 4a–c, the nanowire only grew uniformly in length by 132 nm, while its width stayed unchanged at 62 nm. To maintain uniform growth, two requirements need to be satisfied. On one hand, the neck has to sit near the geometrical center of the nanowire cross section, thus a steady manipulation is vital. On the other hand, it needs some barrier mechanism to prevent incoming atoms from flowing over to the side facets. In fact, at the edges between bottom facets and its vicinal side facets, the diffusing atoms encounter an Ehrlich–Schwoebel barrier[45,46], and this energy barrier at the edge can act in a way similar to a dam to enclose the atomic re-deposition within the end facets. But when a leftward shift of the tip shears the neck by about 25 nm as indicated by the double-harpoon marking in Fig. 4c, the uniform growth in length was interrupted. The facet at the right side instantly started growing. As a result, the apparent diameter of the nanowire increased to 92 nm, as shown in Fig. 4d. To better understand the relation between neck shear and thickening of the

nanowire, the displacement of the tip in the pulling direction ($Y_{tip}$) and lateral direction ($X_{tip}$), as well as the apparent diameter of the nanowire ($\Phi$), are measured digitally and plotted versus the same time axis in Fig. 4e. From the plot, the neck shears at 58 s, followed immediately by a 15 s (from 58 to 73 s) period of the increase of the apparent diameter. The thickening rate is measured to be 1.9 nm s$^{-1}$. In the growth after 73 s, the apparent diameter stayed unchanged at about 92 nm, and the growing velocity is nearly constant at 4.0 nm s$^{-1}$. These results indicate that shearing the neck region may lead to growth of the facet on the opposite side to the shear direction. After the shear, both the diameter and the pulling speed increase, implying dramatic rise in growth rate and the thin neck can still survive. Simple calculation shows that the atomic flux density $J_s$ according to Eq. (1) almost tripled from ~1.9 × 10$^{22}$ m$^{-2}$ s$^{-1}$ between 20 and 58 s to ~7.1 × 10$^{22}$ m$^{-2}$ s$^{-1}$ between 73 and 110 s. By tracking the size at the thinnest part of the neck as shown in Supplementary Fig. 12, we found that the apparent diameter of the neck decreases from 32 to 24 nm. The nanowire adapts this way, because thinner diameter reduces the distance of GB transport and leads to faster deposition.

**Atomistic simulation of thermomigration**. To clarify the atomic motion at the neck region with GB during the thermomigration process, direct molecular dynamics simulations with temperature gradient were performed with and without pulling load (Fig. 5a). We measured atomic displacement and time-averaged mean square atomic displacement (see Supplementary Note 9), such as atomic diffusivity, of each atom during the simulations. Figure 5b shows the atomic displacement vector of each atom during the simulation. Long-distance displacement/diffusion primarily occurs on the neck region surface and substrate surface, and even in the GB. Figure 5c shows the axial component (direction along temperature gradient) of the displacement vector. Under the both loading conditions, atoms on the surface diffuse from high temperature to low temperature region, and some of them diffuse into the GB. The atomic diffusivity of each atom was also calculated (see Supplementary Fig. 13). The atomic diffusivity in the GB is relatively lower than on the surface, i.e., GB plating process

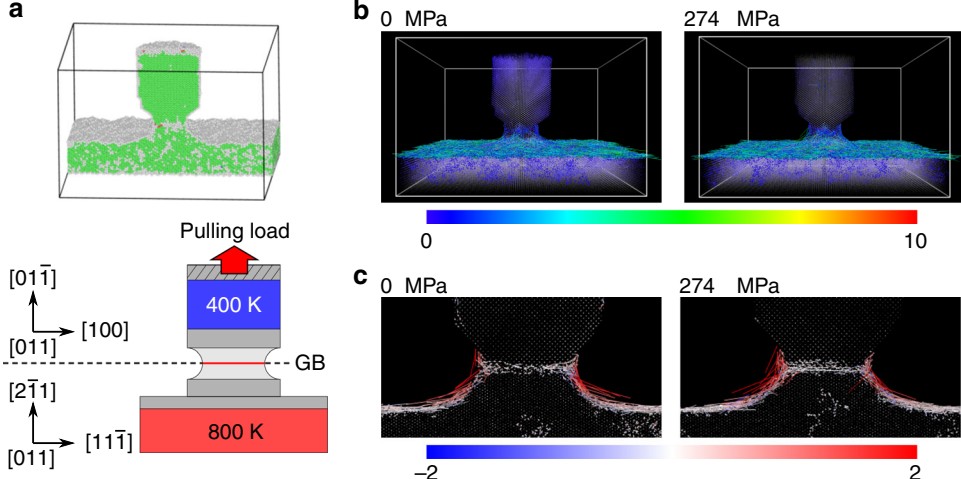

**Fig. 5** Molecular dynamics (MD) simulation of atomic diffusion on an Al nanowire surface. **a** (Top) Simulation supercell of necking nanowire with asymmetric tilt grain boundary (GB) on the substrate. Atoms are colored by common neighbor analysis. The green, red, white atoms indicate the face-centered cubic structure, hexagonal close-packed structure, unknown coordination structure, respectively. (Bottom) Schematic diagram of the simulation with temperature gradient. The height of the intermediate region between red and blue regions is 6.3 nm. **b** Atomic displacement vector in the unit of nm after 10 ns MD simulation. The atomic displacement vector is colored by the magnitude of displacement. **c** Cross-section view of atomic displacement vector. The atomic displacement vector is colored by axial component. Red and blue arrows indicate the atoms moving upward and downward. Length of displacement vectors is reduced by half of the actual length. The MD results were visualized using OVITO[50]

is sluggish. In addition, from Fig. 5c, the pulling load promotes the atomic diffusion both on the surface and in the GB.

## Discussion

We have shown a method of die-free drawing from hot solids to prepare metallic nanowires. The nanowire formation is mediated by thermomigration along surface and then diffusive plating into GB, rather than by displacive deformation, so that single crystal nanowires without internal defects can be obtained. Between the hot end and the nanowire, a neck-shaped ligament usually forms, in which steep temperature gradient exists to incur thermomigration, such that a high atomic flux flows along the neck surface towards the cold side to feed the nanowire growth. The neck-shaped ligament containing a grain boundary is quite robust and its existence is crucial for stable and uniform growth of the nanowire. By manipulating the drawing velocity and direction, the apparent diameter and shape of the nanowire are controllable. The underlying physics is applicable to the preparation of other nanostructures.

Compared with existing methods of nanowire synthesis, this method has advantages in making high-aspect-ratio free-standing nanostructure in a pristine and controlled way, which is a long-standing challenge. For example, photolithography can only fabricate in-plane patterned structure; nano-imprinting usually requires a hard mold to produce free-standing straight nanowire array, but the subsequent de-molding process can often destroy or contaminate the nanowire array. Our method also opens a new avenue to engineering/constructing nanostructures. For example, by using a movable metal tip as the hot source, i.e., a thermo-stylus, arbitrary structures can be deposited on cold substrate, in a way similar to 3D printing. Such additively manufactured 3D nanostructures can be directly used as single-nanowire probes[47], interconnects for repair or connection, or architected nanowire frameworks.

## Methods

**Sample preparation**. Single crystal aluminum (99.9995%) was cut into 1.5 mm × 2 mm × 0.5 mm rectangular plates. The plates were mechanically polished to a thickness of ~100 μm and then one edge was further electrochemically thinned to a few microns in thickness. The aluminum plate was attached onto a MEMS heater mount by conductive epoxy that can survive 500 °C high temperature. On part of the thinned edge, focused ion beam (FIB) was used to reduce the thickness of the front edge to ~2 μm by micromachining.

**Nanowire growth by hot drawing**. Our in situ hot-drawing experiments were carried out with a Hysitron PI95 H1H Picoindenter holder, which was used in a Hitachi H-9500 Transmission Electron Microscope operated in high vacuum (<4 × $10^{-4}$ Pa). The MEMS heater mount, where the aluminum foil sample was attached, was screwed onto the picoindenter holder, which can feed electrical current for heating up the sample. The MEMS heater mount allows a maximum heating temperature of 400 °C, and real-time temperature monitor can be achieved with software control of an embedded resistance temperature detector. The cold end is a tungsten wire with tip radius <100 nm. The tungsten tip is movable, controlled with a two-stage positioning system consisting of piezoelectric precision positioning and mechanical knob coarse positioning.

To grow a nanowire, the aluminum plate sample was firstly heated up to 400 °C (~$0.72T_m$ of aluminum) with the heating rate of ~0.3 °C s$^{-1}$. The cold tungsten tip was manipulated to touch the edge of hot aluminum foil to expose fresh metal surface by breaking the native oxide layer. Then the cold tungsten tip was pulled backward in a stepped manner with a speed of a few nanometers per second. The nanowire growth process was observed and recorded by Gatan OneView camera at 5 frames per second. The tip movement was retrieved from the video by tracking a selected feature on the nanowire or tip with the commercial software of Adobe After Affect. The output is a trajectory in pixel coordinates, which can be converted into nanometers using scale bar, thus the instant velocity at any given time can be obtained.

**Molecular dynamics simulation**. Direct molecular dynamics simulations with temperature gradient were performed using Large-scale Atomic/Molecular Massively Parallel Simulator (LAMMPS) packages[48]. An Al nanowire model with necking ($\Phi_w$ = 8 nm, $\Phi_n$ = 5 nm) consist of 187,102 atoms was used, which contains tilt grain boundary as shown in Fig. 5a. The embedded atom method (EAM) potential[49] was used for interatomic interaction. The regions above necking and

below the substrate surface were maintained constant temperature of 127 °C (400 K) and 527 °C (800 K), respectively. Thus the temperature gradient at necking region was estimated at ~6.3 × $10^{10}$ K/m. Ten nanoseconds of MD simulations were run with and without pulling load along axial direction. The applied load corresponds to nanowire pulling stress of 274 MPa.

**Reporting summary**. Further information on research design is available in the Nature Research Reporting Summary linked to this article.

## Data availability
The data that support the findings of this study are available from the corresponding authors on request.

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

## Acknowledgements

The authors acknowledge supports from the National Key Research and Development Program of China (2017YFB0702001), Natural Science Foundation of China (51701151), 111 Project 2.0 (BP2018008), Shaanxi Postdoctoral Science Foundation (2017JQ5110). S. O. acknowledges the support by JSPS KAKENHI (Grant Nos. JP18H05450, JP18H05453, JP17H01238, JP17K18827) and Elements Strategy Initiative for Structural Materials (ESISM). J.L. acknowledges support by NSF DMR-1410636. E.M. acknowledges support from U.S. DoE-BES-DMSE, under Contract No. DE-FG02-16ER46056.

## Author contributions

J.L., and Z.S. designed the project. D.X., Z.N., and Y. Y. conducted the experimental work. S.S. conducted MD simulation. S.O. guided the FEM analysis, MD simulation and the theoretical understanding. D.X., E.M., and J.L. wrote the paper. All authors contributed to the discussions of the results.

## Competing interests

The authors declare no competing interests.
