## [Peer Review File · Nature Communications]

Reviewers' comments:

Reviewer #1 (Remarks to the Author):

The manuscript describes the growth of aluminum nanowires across a junction of thermal gradient inside a TEM. The authors claimed that this phenomenon was due to the thermomigration of atoms between substrates at different temperatures. They applied the classical FEM method and claimed that some unusual mechanical back-stress must involve in the process because a huge temperature gradient is theoretically predicted for this process to happen. The reviewer is not sure about the parameters used in the FEM modeling. However, it seems that the model does not include surface tension factors. It is well known that the melting points of metals may decrease when their sizes decrease to 10s or a few nm. A commonly cited example is the melting point of very small gold spheres. The surfaces of them can be depressed from over 1000C to a few hundreds. The authors should address this factor. Perhaps, a semi-classical method could better assist the modeling of this system. It will be helpful if the authors can address the effect of electron beam in the process in this manuscript.

Overall, the demonstrate experimental methods are straight forward and are interesting to the nanowire community. However, the significance of using this method for mass production of nanowires or other metal nanowire systems unclear. The authors can increase the significance of the method by demonstrating its applicability to other metal or alloy metal systems.

Reviewer #2 (Remarks to the Author):

Comments on the manuscript : "controlled growth of aluminum nanowire via thermomigration across a nanoscaled junction" by Xie DG et al. (Nature Communication)

This paper demonstrated a robust and controlled new method to prepare single-crystalline metallic nanowires (of nearly uniform diameter typically around a few tens of nanometers) by drawing a cold nano-tip from a hot solid based on temperature-gradient driven thermomigration. The demonstrated new method combines the traditional method of growing single-crystalline out of a liquid and the known metalworking techniques of wire drawing. The experimental results shown in this paper clearly confirmed that the temperature gradient from hot solid to cold nanotip can reach a level much higher than in bulk body, and the nanowire formation is certainly dominated by temperature-gradient driven thermal diffusion and is well controllable by manipulating the drawing velocity and direction. The produced nanowires are demonstrated to sustain uniaxial tensile stress. The problem studied here is of great interest and significant application to nanowire technology. In particular, compared with existing methods of nanowire synthesis, this interesting method offers a hopeful new approach to make high-aspect ratio free-standing nanowires of nearly uniform diameter in a clean and controlled way. The manuscript is well written with clearly identified preconditions of the proposed new method, and very likely has the potential to stimulate wide interest and make a significant impact to the techniques of nanowire formation. Without a hesitance, I recommend it for timely publication in Nature Communication.

A suggestion for an optional revision is that the present version shows only 3 figures of experimental results. I believe that adding more relevant experimental figures can make the paper even more attractive and interesting.

Reviewer #3 (Remarks to the Author):

Review Questions:

What are the major claims of the paper?

The authors claim to have developed a technique to fabricate Al (a highly reactive material) nanowires using thermomigration. This technique produces single crystal and defect tolerant nanowires.

Are they novel and will they be of interest to others in the community and the wider field?

The study of electromigration and thermomigration has been acknowledged by the authors as a studied problem in integrated circuits. Here, they take a known problem and have flipped it to fabricate a nanowire. Here, the novelty would be to understand the physics of formation and diffusion within crystalline nanowires by focusing on different surfaces, defects, sizes at the nanoscale (using the TEM).

Is the work convincing, and if not, what further evidence would be required to strengthen the conclusions?

A general comment, this paper should be more carefully copyedited to fix grammatical issues.

Authors:

Lines 59-61: For integrated circuits, thermomigration has been considered as a threat to structural reliability of nanoscale interconnects^{7,8} with occasional temperature gradient typically < 1000 °C/cm.

Reviewer Comment:

Some key points of the analysis that are missing have to do with an in depth discussion on crystallography, measurements of grain sizes, defects of the structures produced and any comment on dominant orientation of the crystallography and whether anisotropy has any effect in presenting their analysis and model. Given the uncertainties in the experiment (such as quantitative measurements of thermal gradients or including molecular dynamic based models that can be used to predict the gradient at the appropriate scales) there should be a deeper discussion on diffusion based on the orientation of crystal grains, defect types and density of defects. The analysis relies on bulk properties of surface diffusion or thermal gradient analysis (i.e. using FEM), which limits the ability to translate the results to other nanoscaled materials or have the results repeated by other researchers. In addition, because the analysis does not use more detailed that are compatible to the scales of the experimental structures, such as molecular dynamics, the accuracy of the results and their analysis can have large errors. Note, the reviewers do not address this crystallography discussion until lines 205-216 and state that the wire is "single crystal" with a "pristine interior." Is this true for all wires as the surfaces change, such as having a zig-zag surface, based on the angle used to pull the wire? In this reviewer's opinion, performing detailed MD analysis for the different types of nanowires would result in deeper understanding of the thermomigration in these structures and result in broader application by the materials or engineering community.

Authors:

Lines 147-148: Fig. 2e to 2f show the nanowire in the uniform growth stage, with a nearly constant velocity of $v=3.9$ nm/s.

Review comment: How did you arrive to this measurement? This should also have uncertainty associated with it that should be included.

Lines 183-186: The huge ∇T gap between the requirement for thermomigration and FEM results indicates that there must exist some backward process to counter-balance the thermomigration, such as mass back flow driven by back stress, a phenomenon well noticed as a side effect during thermomigration in interconnects of microchips.

Review comment: As mentioned above, this reviewer has doubts on the applicability of using FEM and treating the nanowire as an isotropic material in the analysis. Per the discussion by the authors below, they are dealing with crystals not an isotropic material.

Authors:

Lines 205-216: The surface morphology of the nanowire can be controlled by changing the pulling direction, as shown in Figure 3 and supplementary movie S2 and S3. The initial pulling direction is about 26° away from a $\langle 110 \rangle$ direction of the nanowire, and the obtained nanowire developed a saw-tooth-shaped surface, as shown in Figure 3a-c and movie S2. However, when the pulling direction is $\sim 4^\circ$ away from the same $\langle 110 \rangle$ direction, the nanowire can almost maintain a straight and smooth morphology, except for a few small surface steps, as shown in Fig. 3d-e and movie S3. Thus, it can be concluded that pulling along a low-index direction of the nanowire tends to produce a straight prismatic nanowire. Fig 4f is the dark field image of the nanowire taken with $[1\bar{1}\bar{1}]$ diffraction spot, showing that the whole nanowire lights up and clear thickness fringes appears. These two signs indicate that despite the switch of pulling direction, the final nanowire still maintains a single crystal with pristine interior.

Authors:

Lines 231-232: (check textbook on phase transformations and <http://li.mit.edu/S/PM>).

Reviewer comment: This is not a proper citation.

Authors:

Line 270: growing velocity is nearly constant at 4.0 nm/s.

Reviewer comment: Again, how was this measured and what is the uncertainty?

Authors:

Lines 298-301: As an example, by using a movable metal tip as the hot source, i.e., a “thermo-stylus”, arbitrary structures can be deposited on cold substrate, in a way similar to 3D printing. Such additively manufactured 3D nanostructures can be directly used as single-nanowire probes⁴¹, interconnects for repair or connection, or architected nanowire frameworks.

Reviewer comment: The technique presented in this paper has very limited scalability. They are presenting high vacuum fabrication of Al nanowires without oxide, which cannot be preserved unless they cap it before removing from the UHV environment. Although 3D printing is limited by the ability to draw the nanowires in a “dirty” environment which would result in wires that have contaminants and likely have more defects. The packing densities of the nanowires would also will likely be limited and it is doubtful that high packing densities or accurate placements could be achieved (such as that needed for interconnect repairs).

Authors:

In the supplementary material section 2, Estimation of the pulling stress inside the neck-shaped ligament

Reviewer comment on reproducibility: What is the calibration curve for the “high-sensitivity force sensor (≤ 3 nN).” used in to or the detailed load/unload cycling curves to account for possible hysteresis if there is any.

On a more subjective note, do you feel that the paper will influence thinking in the field?

Reviewer comment: The framing of the results to an isotropic material ignores the impact that crystallinity has on the properties of nanoscale materials where they can significantly affect the material properties. Because of the experimental uncertainties, trying to say that this can be scaled to devices using 3D printing techniques is also questionable. Thus, this limits the ability of this paper to influence the broader community in either materials design, fabrication or application.

-Reviewer #1 (Remarks to the Author)

RIQI: *The manuscript describes the growth of aluminum nanowires across a junction of thermal gradient inside a TEM. The authors claimed that this phenomenon was due to the thermomigration of atoms between substrates at different temperatures. They applied the classical FEM method and claimed that some unusual mechanical back-stress must involve in the process because a huge temperature gradient is theoretically predicted for this process to happen. **The reviewer is not sure about the parameters used in the FEM modeling.** However, it seems that the **model does not include surface tension factors.** It is well known that the **melting points of metals may decrease when their sizes decrease to 10s or a few nm.** A commonly cited example is the melting point of very small gold spheres. The surfaces of them can be depressed **from over 1000C to a few hundreds.** The authors should address this factor. Perhaps, **a semi-classical method** could better assist the modeling of this system.*

Reply: We thank the reviewer for pointing out the possible impact that surface tension may have on the FEM modeling.

It's true that the surface tension effect can depress the melting point, especially when the sample size is below ten nanometers. As shown in Figure R1, we have estimated this effect using a semi-empirical function from Qi W H. [J]. *Physica B: Condensed Matter*, 2005, 368. The melting point of the 7 nm neck (smallest dimension among all the experiments in our work) is about 0.9 of the bulk value, still far above our experimental temperature ($0.72 T_m$). And also as stated in Supplementary Note 2, the shear strength of the neck is estimated to be ~300 MPa, indicating that **the neck is still strong solid.** In the FEM modeling, a simplified static heat transfer problem is solved to estimate the temperature distribution. For a solid material at small scale, the heat transfer process may be influenced by the surface tension in two aspects, (1) influencing the thermal parameters, including the thermal conductivity and the heat capacity, particularly close to the free surface; and (2) changing the geometric profile of the neck area.

Since the influence on the thermal parameters caused by free surface is lacking in the reported literature according to the best of our knowledge, in the present FEM

modelling, thermal parameters are obtained based on available data and listed as,

- thermal conductivity is derived based on a piecewise constant function according to the width of nanowire d (see Supplementary Fig. 9)

$$k = \begin{cases} 40 \text{ W}/(\text{m} \cdot \text{K}), & \text{when } d=7\text{nm} \\ 75 \text{ W}/(\text{m} \cdot \text{K}), & \text{when } d=40\text{nm} \end{cases}$$

- heat capacity for bulk material is used.

On the other hand, geometric profile of the FEM model is developed according to the experimental observation, where the profile of the neck area is stable during the growth of the nanowire (surface tension may play an important role in maintaining the profile, which is yet another interesting physical problem worth further discussion elsewhere). Therefore, for a heat transfer process of a solid with a given geometric profile, contribution from surface tension is negligible.

Figure R1. The semi-classical relationship between size of aluminum nanowire and melting temperature (Qi W H. [J]. Physica B: Condensed Matter, 2005, 368(1-4): 46-50.)

R1Q2: It will be helpful if the authors can address the effect of electron beam in the process in this manuscript.

Reply: The reviewer raised a very good question, i.e. the beam effect for experiments inside TEM. We have tried to grow nanowire with the electron beam turned off, and the results showed that nanowire growth wasn't interrupted, as shown in Figure R2. The

nanowire segments with and without beam illumination showed no obvious difference. This is evidence that electron beam effect is negligible to the reported phenomenon under our experimental condition.

This experimental result is also added as Supplementary Note 4 and Supplementary Figure 5.

Figure R2. A nanowire grown under both beam-ON and beam-OFF conditions. The segments grown under beam-OFF condition are labeled with dashed red rectangle.

R1Q3: Overall, the demonstrate experimental methods are straight forward and are interesting to the nanowire community. However, the significance of using this method for mass production of nanowires or other metal nanowire systems unclear. The authors can increase the significance of the method by demonstrating its applicability to other metal or alloy metal systems.

Reply: The transferability to other metal systems is also our concern. To address this concern, we have experimented with other metals including Cu, Ag and Sn, as shown in Figure R3, R4 and R5. These results unambiguously demonstrate the general applicability of our new method. Actually, this generality is rooted in thermomigration, which is a universal physical phenomenon driven by temperature gradient.

These results were also added to the supplementary materials as Supplementary Fig. 1-3 and Supplementary Note 1.

Figure R3. The growth of a copper nanowire via thermomigration with exact experimental set-up. The Cu matrix is hold at 400 °C ($\sim 0.50T_{m,Cu}$). All the scale bars are

100 nm.

Figure R4. The snapshots of silver nanowires grown from Ag matrix at 400 °C ($\sim 0.55T_{m,Ag}$). The scale bar in (a) is 50 nm. Scale bars in (b)-(e) represent 200 nm.

Figure R5. The making of a Sn nanowire by drawing W tip at RT from the Sn substrate at 120 °C ($\sim 0.78 T_{m,Sn}$).

-Reviewer #2 (Remarks to the Author)

R2Q1: This paper demonstrated a robust and controlled new method to prepare single-crystalline metallic nanowires (of nearly uniform diameter typically around a few tens of nanometers) by drawing a cold nano-tip from a hot solid based on temperature-gradient driven thermomigration. The demonstrated new method combines the traditional method of growing single-crystalline out of a liquid and the known

*metalworking techniques of wire drawing. The experimental results shown in this paper clearly confirmed that the temperature gradient from hot solid to cold nano-tip can reach a level much higher than in bulk body, and the nanowire formation is certainly dominated by temperature-gradient driven thermal diffusion and is well controllable by manipulating the drawing velocity and direction. The produced nanowires are demonstrated to sustain uniaxial tensile stress. The problem studied here is of great interest and significant application to nanowire technology. In particular, compared with existing methods of nanowire synthesis, this interesting method offers **a hopeful new approach to make high-aspect ratio free-standing nanowires of nearly uniform diameter in a clean and controlled way.** The manuscript is well written with clearly identified preconditions of the proposed new method, and very likely **has the potential to stimulate wide interest and make a significant impact to the techniques of nanowire formation.** Without a hesitation, I recommend it for timely publication in Nature Communication.*

*A suggestion for an optional revision is that the present version shows only 3 figures of experimental results. I believe that **adding more relevant experimental figures** can make the paper even more attractive and interesting.*

Reply: We thank the reviewer for recapitulating the key points and pointing out the significance of our finding for nanowire technology.

We further tried similar experiments on Cu, Ag and Sn, as shown in Figure R3, R4, and R5. These results have been added into the main article and the supplementary materials (Supplementary Note 1 and Supplementary Fig. 1-3).

We also performed atomistic simulations of the thermomigration process in the neck region, which has been added to the main article as Figure 5.

-Reviewer #3 (Remarks to the Author)

R3Q1:Review Questions:

What are the major claims of the paper?

The authors claim to have developed a technique to fabricate Al (a highly reactive material) nanowires using thermomigration. This technique produces single crystal and defect tolerant nanowires.

Reply: Thank reviewer for the concise summary of our claims.

R3Q2:Are they novel and will they be of interest to others in the community and the wider field?

*The study of electromigration and thermomigration has been acknowledged by the authors as a studied problem in integrated circuits. **Here, they take a known problem and have flipped it to fabricate a nanowire.** Here, the novelty would be to understand the physics of formation and diffusion within crystalline nanowires buy focusing on different surfaces, defects, sizes at the nanoscale (using the TEM).*

Reply: We are sorry for not making the novelty of this work clear enough in the article. The novelty is that the single nanoscale junction between two bulk-scale heat source and sink can create super high temperature gradient, estimated at the scale of $\sim 10^9$ K/m, the highest value ever made use of, to the best of our knowledge. Without this super high temperature gradient, witnessing the thermal migration in the experimental timescale would be impossible. This super-high temperature gradient imparts the capability of rapid shape change to small-scaled structure. The good controllability even with our simple experimental setup demonstrates the potential of maneuvering thermomigration for applications in broader nanotechnologies.

R3Q3:Is the work convincing, and if not, what further evidence would be required to strengthen the conclusions?

*A general comment, this paper should be **more carefully copyedited to fix grammatical issues.***

Reply: We thank the reviewer for the suggestion. The grammatical mistakes in the article have been corrected..

R3Q4:Authors:

Lines 59-61: For integrated circuits, thermomigration has been considered as a threat to structural reliability of nanoscale interconnects^{7,8} with occasional temperature gradient typically < 1000 °C/cm.

Reviewer Comment:

*Some key points of the analysis that are missing have to do with an **in depth discussion on crystallography, measurements of grain sizes, defects of the structures** produced and any comment on dominant orientation of the crystallography and **whether anisotropy has any effect** in presenting their analysis and model.*

*Given the uncertainties in the experiment (such as quantitative measurements of thermal gradients or including molecular dynamic based models that can be used to predict the gradient at the appropriate scales) there should be a deeper discussion on diffusion based on the **orientation of crystal grains, defect types and density of defects**. The analysis relies on bulk properties of surface diffusion or thermal gradient analysis (i.e. using FEM), which limits the ability to **translate the results to other nanoscaled materials or have the results repeated by other researchers**. In addition, because the analysis does not use more detailed that are compatible to the scales of the experimental structures, such as **molecular dynamics**, the accuracy of the results and their analysis can have large errors.*

*Note, the reviewers do not address this crystallography discussion until lines 205-216 and state that **the wire is “single crystal” with a “pristine interior.”** Is this true for all wires as the surfaces change, such as having a zig-zag surface, based on the angle used to pull the wire? In this reviewer’s opinion, performing detailed MD analysis for the different types of nanowires would result in deeper understanding of the thermomigration in these structures and result in broader application by the materials or engineering community.*

Reply: We thank the reviewer for pointing out these missed points and the suggestion of performing MD analysis.

All produced Al nanowires can be identified to have $\langle 110 \rangle$ axial direction, as shown in

Figure R6. The apparent diameters of uniformly-grown nanowires ranges from 7 nm to 320 nm. The crystallographic orientation of the nanowire is mainly determined by the minimization of total surface energy, which always results in a $\langle 110 \rangle$ axis. The substrate does not seem to have obvious effect, since redeposition at the neck region happens at the surface of cold side or inside the grain boundary, and the newly deposited atoms comply with the orientation of the nanowire, which may be initially seeded by the cold tip.

Figure R6. Typical examples showing produced aluminum nanowires display axial orientation of $\langle 011 \rangle$. All scale bars represent 200 nm.

We have added more discussions/comments in the main article to address the effect of crystallography and grain size.

We admit that only part of the nanowires is single crystalline, when the drawing motion is stable without substantial vibration. The movable tip in our experiment always comes with some vibration, resulting from the mechanical or electromagnetic noise in the system. Any substantial vibration will introduce sporadic surface imperfections to the nanowire. It is however anticipated that perfect single crystalline nanowire can be

obtained with a drawing tip of better steadiness.

The nanowire in Figure 3 is “single crystal” with a “pristine interior”, for the following reasons: 1) The whole nanowire shows only one set of diffraction pattern, and lights up in dark-field imaging using one diffracted spot. 2) The smooth thickness fringes were observed across the whole nanowire, including the straight part and the zig-zag part. Thickness contour is very sensitive to lattice defects and surface shape. The smooth fringes in Figure 3.f indicates a nearly pristine interior. 3) No grain boundary was found in any of the bright/dark field images taken from different tilt angles. These demonstrate that the nanowire is a single crystal with a very clean interior.

MD analysis is not good at predicting the thermal conductivity and therefore cannot be used to determine the real thermal gradient in the nanowire. However, we do use MD to simulate the atomic processes at the neck with an grain boundary, under uniform temperature distribution and with/without tensile stress, to see the diffusion of surface atoms into the grain boundary. This part is added to the revised article as an independent section and Figure 5.

R3Q5:Authors:

Lines 147-148: Fig. 2e to 2f show the nanowire in the uniform growth stage, with a nearly constant velocity of $v=3.9$ nm/s.

*Review comment: **How did you arrive to this measurement?** This should also have uncertainty associated with it that should be included.*

Reply: We have used commercial software suites of Adobe After Affect to track the motion of a selected feature in the tip or nanowire, by analyzing the video frame by frame. The obtained trajectory of the feature is given in pixel coordinates, which can be converted into nanometers using scale bar, thus the instant velocity at any given time can be obtained. This information has been added to the Methods part in the main article. The measurement of the $v=3.9$ nm/s was supported by the original data in Supplementary Figure 2, which was copied to Figure R7 below.

Figure R7. The measurement of the drawing velocity in Figure 2.

Since the measurement has a resolution of one pixel, the uncertainty of the measured coordinates is less than one pixel (1 pixel \approx 1.15 nm/pixel), which is \sim 1% of the total traveling distance. Therefore, we only kept 2 significant digits for velocity values.

R3Q6: Lines 183-186: The huge ∇T gap between the requirement for thermomigration and FEM results indicates that there must exist some backward process to counterbalance the thermomigration, such as mass back flow driven by back stress, a phenomenon well noticed as a side effect during thermomigration in interconnects of microchips.

Review comment: As mentioned above, this reviewer has doubts on the applicability of using FEM and treating the nanowire as an isotropic material in the analysis. Per the discussion by the authors below, they are dealing with crystals not an isotropic material.

Reply: Since the FEM modeling is only used to estimate the temperature distribution in the neck region, we guess that the reviewer's concern is that the orientation-dependent thermal parameters shall be used for correct output. Unfortunately, we

searched the literature and found no reported data on these parameters for nanoscale aluminum crystal. Therefore, we only use FEM modeling to roughly estimate the orders of magnitude of temperature gradient in the neck region.

For anisotropic materials such as Sn, we can grow a nanowire using the same experimental setup, as shown in Figure R5.

R3Q7:Authors:

Lines 205-216: The surface morphology of the nanowire can be controlled by changing the pulling direction, as shown in Figure 3 and supplementary movie S2 and S3. The initial pulling direction is about 26° away from a $\langle 110 \rangle$ direction of the nanowire, and the obtained nanowire developed a saw-tooth-shaped surface, as shown in Figure 3a-c and movie S2. However, when the pulling direction is $\sim 4^\circ$ away from the same $\langle 110 \rangle$ direction, the nanowire can almost maintain a straight and smooth morphology, except for a few small surface steps, as shown in Fig. 3d-e and movie S3. Thus, it can be concluded that pulling along a low-index direction of the nanowire tends to produce a straight prismatic nanowire. Fig 4f is the dark field image of the nanowire taken with $[1\bar{1}\bar{1}]$ diffraction spot, showing that the whole nanowire lights up and clear thickness fringes appears. These two signs indicate that despite the switch of pulling direction, the final nanowire still maintains a single crystal with pristine interior.

Authors:

Lines 231-232: (check textbook on phase transformations and <http://li.mit.edu/S/PM>).

*Reviewer comment: This is **not a proper citation**.*

Reply:

Thanks for the pointing out this improper citation. We have replaced this with the following two articles:

- [1] Ford, J. M., Wheeler, J. & Movchan, A. B. Computer simulation of grain-boundary diffusion creep. Acta Mater. 50, 3941-3955 (2002).
- [2] Zhang, W. & Schneibel, J. H. The sintering of 2 particles by surface and grain-boundary diffusion - A 2-dimensional numerical study. Acta Metall. Mater. 43, 4377-

4386 (1995).

R3Q8:Authors:

Line 270: growing velocity is nearly constant at 4.0 nm/s.

Reviewer comment: Again, how was this measured and what is the uncertainty?

Reply:

This was measured using the same methods mentioned above in reply to R3-Q5.

R3Q9:Authors:

Lines 298-301: As an example, by using a movable metal tip as the hot source, i.e., a “thermo-stylus”, arbitrary structures can be deposited on cold substrate, in a way similar to 3D printing. Such additively manufactured 3D nanostructures can be directly used as single-nanowire probes⁴¹, interconnects for repair or connection, or architected nanowire frameworks.

*Reviewer comment: The technique presented in this paper has **very limited scalability**. They are presenting high vacuum fabrication of Al nanowires without oxide, which cannot be preserved unless they cap it before removing from the UHV environment. Although 3D printing is limited by the ability to draw the nanowires in a “dirty” environment which would result in wires that have contaminants and likely have more defects. The packing densities of the nanowires would also will likely be limited and **it is doubtful that high packing densities or accurate placements could be achieved** (such as that needed for interconnect repairs).*

Reply: The reviewer mentioned a few concerns about the scalability. We insist that our method has a good potential to scale-up. The scalability of our method depends on the stability of the instrument. The state-of-the-art equipment in the semiconductor industry can actually achieve much higher positioning resolution (better than 0.15 nm in ASML lithography system) than our simple experimental instrument. The vacuum chamber

can also be replaced with a chamber filled with inert gas or reducing gas, so that oxidation of fresh nanowires can be effectively prevented.

R3Q10:Authors:

In the supplementary material section 2, Estimation of the pulling stress inside the neck-shaped ligament

Reviewer comment on reproducibility: What is the calibration curve for the “high-sensitivity force sensor (≤ 3 nN).” used in to or the detailed load/unload cycling curves to account for possible hysteresis if there is any.

Reply: We used the Hysitron PI95 Picoindenter to probe the force inside TEM. The product brochure claimed a load force resolution ≤ 3 nN and load noise floor < 0.2 μ N. In real applications, the actual load noise floor and force resolution depend on the noise level in the environment, but still enough to resolve the uN level load.

We replaced the “(≤ 3 nN)” in the main article with “(Hysitron PI95 Picoindenter)”, which could be more appropriate.

R3Q11:On a more subjective note, do you feel that the paper will influence thinking in the field?

Reviewer comment: The framing of the results to an isotropic material ignores the impact that crystallinity has on the properties of nanoscale materials where they can significantly affect the material properties. Because of the experimental uncertainties, trying to say that this can be scaled to devices using 3D printing techniques is also questionable. Thus, this limits the ability of this paper to influence the broader community in either materials design, fabrication or application.

Reply: If a more stable instrument is developed, site-specific growth/repair in a super clean way is still possible. The authors agree that instrumental development is not a simple task, maybe very challenging. However, there is still a good chance with available technology in the market.

The significance of this work also lies in the good transferability, which was experimentally demonstrated with Cu (Figure R3), Ag (Figure R4), and Sn (Figure R5).

REVIEWERS' COMMENTS:

Reviewer #1 (Remarks to the Author):

The authors sufficiently addressed the reviewers' questions in the revised manuscript. Therefore, the reviewer recommends the manuscript for publication.

Reviewer #3 (Remarks to the Author):

This reviewer would like to thank the authors for their additional work to address concerns in the original manuscript.

The additional results showing fabrication of nanowires with other materials strongly supports the conclusion that the technique is transferrable to other materials and can be used for directed growth of nanowires. The additional modeling and analysis to elucidate the mechanics of the growth also supports the authors' analysis and addresses earlier concerns.

Although this reviewer respectfully disagrees on scalability of the method as a large scale commercial technique, the reviewer agrees that there can be unique applications of the fabrication method for future applications. Thus, the modifications to the manuscript satisfies the concerns of this reviewer.